# Appraising the Role of Astrocytes as Suppliers of Neuronal Glutathione Precursors

**DOI:** 10.3390/ijms24098059

**Published:** 2023-04-29

**Authors:** Dolores Pérez-Sala, María A. Pajares

**Affiliations:** Department of Structural and Chemical Biology, Centro de Investigaciones Biológicas Margarita Salas (CSIC), Ramiro de Maeztu 9, 28040 Madrid, Spain

**Keywords:** glutathione, astrocyte and neuron communication, amino acid transport, neurodegenerative diseases, Alexander disease, extracellular vesicles, amino acid metabolism

## Abstract

The metabolism and intercellular transfer of glutathione or its precursors may play an important role in cellular defense against oxidative stress, a common hallmark of neurodegeneration. In the 1990s, several studies in the Neurobiology field led to the widely accepted notion that astrocytes produce large amounts of glutathione that serve to feed neurons with precursors for glutathione synthesis. This assumption has important implications for health and disease since a reduction in this supply from astrocytes could compromise the capacity of neurons to cope with oxidative stress. However, at first glance, this shuttling would imply a large energy expenditure to get to the same point in a nearby cell. Thus, are there additional underlying reasons for this expensive mechanism? Are neurons unable to import and/or synthesize the three non-essential amino acids that are the glutathione building blocks? The rather oxidizing extracellular environment favors the presence of cysteine (Cys) as cystine (Cis), less favorable for neuronal import. Therefore, it has also been proposed that astrocytic GSH efflux could induce a change in the redox status of the extracellular space nearby the neurons, locally lowering the Cis/Cys ratio. This astrocytic glutathione release would also increase their demand for precursors, stimulating Cis uptake, which these cells can import, further impacting the local decline of the Cis/Cys ratio, in turn, contributing to a more reduced extracellular environment and subsequently favoring neuronal Cys import. Here, we revisit the experimental evidence that led to the accepted hypothesis of astrocytes acting as suppliers of neuronal glutathione precursors, considering recent data from the Human Protein Atlas. In addition, we highlight some potential drawbacks of this hypothesis, mainly supported by heterogeneous cellular models. Finally, we outline additional and more cost-efficient possibilities by which astrocytes could support neuronal glutathione levels, including its shuttling in extracellular vesicles.

## 1. Introduction

The brain is a complex organ formed by a variety of highly specialized structures and cell types that consume a large amount of energy for appropriate functioning. In order to meet these energy needs, several studies have estimated that the brain consumes a high proportion of oxygen (nearly 20% [1,2,3]) and glucose (around 25% [4]) required by an organism. Moreover, energy requirements are not uniform in the whole tissue, and areas of high neuronal activity are more demanding. The adequate supply of both oxygen and glucose to different areas entails a tight control of the cerebral blood flow, involving the intercommunication between several cell types. On the one hand, neurons release glutamate (Glu; throughout this manuscript the three-letter code for amino acid abbreviations will be used referring to the L-isomers) and nitric oxide (NO), the latter exerting its vasodilating effects on the arterioles. On the other hand, astrocytes increase their Ca^2+^ concentrations [5,6,7], which have been reported to show vasodilation or vasoconstriction effects and to spread in waves between astrocytes. The combined actions of increased Ca^2+^ levels and decreased oxygen supply, which induce glycolytic production and release of lactate from astrocytes, in turn, lead to prostaglandin E_2_ accumulation and vasodilation [8], which simultaneously correlates with the elevation of extracellular adenosine levels to block vasoconstriction [9,10].

Once the fuels (glucose and oxygen) reach the cells, the production of energy in the form of ATP is achieved through the mitochondrial electron transport chain (ETC) and/or glycolysis. The use of either pathway or both seems to differ between cell types and, in fact, the literature indicates that neurons rely preferentially on the ETC, while astrocytes use glycolysis [11,12]. Reactions catalyzed at complexes I and III of the ETC are the main leaky steps leading to the production of reactive oxygen species (ROS). During normal respiration, 1–2% of the oxygen consumed is converted to superoxide anion and, in turn, to other ROS, but this process can contribute to the generation of additional free radicals, including reactive nitrogen species, carbon- and sulfur-centered radicals [13,14,15]. Low levels of ROS, produced by the ETC and many additional oxygen-consuming reactions, are used as signaling molecules [16]. However, high levels of these free radicals can damage the DNA, proteins, and lipids, and hence to prevent these negative effects, all cell types have a variety of antioxidant response systems [17,18]. Among them, some require trace elements (as cofactors or structural components [19]) that are also important for several aspects of brain function [20,21,22] and which are redox active. The problem arises when the antioxidant response systems become surpassed leading to oxidative stress, a condition to which the brain is especially susceptible [23]. In fact, disruption of oxygen metabolism and mitochondrial function is a hallmark of neurodegenerative diseases [24,25,26], the associated oxidative stress laying the basis for the proteostasis problems, including misfolding and aggregation, commonly found in many of these pathologies (e.g., Parkinson or Alexander disease). 

Antioxidant response systems can be classified into (i) antioxidant enzymatic systems including superoxide dismutase (SOD), glyoxalase, glutathione reductase (GSR), glutathione peroxidases (GPXs), thioredoxin/thioredoxin reductase, and catalase [27], and (ii) low-Mr antioxidants such as glutathione, uric acid, vitamins (A, C, and E), coenzyme Q, and melatonin [28]. Levels of antioxidant enzymatic systems are diverse among brain cell types as reported in the Human Protein Atlas (https://www.proteinatlas.org/ (accessed on 14 November 2022)) and their activities can be impaired by oxidative or electrophilic modifications. Similarly, low-Mr antioxidants also display variations in concentration and distribution. Thus, ascorbate (vitamin C) and glutathione are the most abundant low-Mr antioxidants in the rat central nervous system (CNS), where they display a regional distribution [29]. Importantly, changes in low-Mr antioxidants occur in rats during the first postnatal weeks, which associate with preferential glutathione localization in glial cells compared to neurons (3.8 vs. 2.5 mM) and an opposite pattern for ascorbate distribution (10 vs. 20.9 mM) [29]. Of note, this study also found that ascorbate content in mammals increased with neuronal density, whereas no change in glutathione levels was detected [29]. Studies using primary cultures of newborn (nA) or embryonic astrocytes (eA) and embryonic neurons (eN) also reported some differences regarding the detection and levels of certain low-Mr antioxidants [30,31]. 

Main intracellular low-Mr redox pairs are reduced/oxidized glutathione (GSH/GSSG) and cysteine/cystine (Cys/Cis), while extracellularly and in plasma this role corresponds to the Cys/Cis pair [16,32]. Both redox pairs are intimately linked, as will be discussed later since Cys is among the three amino acids that compose glutathione. Here, we are going to focus our attention on glutathione and its levels and synthesis in neurons and astrocytes, given the crucial importance of this metabolite as ROS scavenger in normal brain physiology and during the oxidative stress associated with neurodegeneration. Special attention will be paid to the hypothesis postulated in the 1990s that suggested a key role of astrocytes as providers of precursors to sustain neuronal glutathione levels. This hypothesis imposes a large expenditure of ATP in the exchange and recycling of components between both cell types, as we shall see in the next sections. For this purpose, some specific characteristics of each cell type need to be described, as well as main aspects of glutathione synthesis and regulation. Aided by the literature and data available in the Human Protein Atlas, concerning expression and protein levels, we are going to discuss the pillars supporting this widely accepted model and their weaknesses.

## 2. General Facts about Glutathione

Glutathione is a special tripeptide synthesized in the cytoplasm of every cell [33], which exists as a reduced (GSH) and oxidized (GSSG) species. Its total cellular concentrations reach between 1 and 10 mM depending on the cell type [34]. Under normal conditions, GSSG represents less than 1% of the total cellular glutathione and this excess of GSH over GSSG is responsible for the reducing potential of the cytoplasm. Therefore, changes in the GSH/GSSG ratio can be used as an indicator of the degree of oxidative stress of a cell or tissue, which in the liver is considered severe if below 50 and mild when the ratio becomes 50–100 [35,36]. Importantly, a comparison of GSH and GSSG concentrations, as well as GSH/GSSG ratios, of the same tissue obtained from different mice strains has shown deviations up to 3-fold [37]. Hence, it should be kept in mind that direct comparisons between data of different studies that may use diverse strains and/or organisms may not be adequate.

Although nearly 90% of GSH is found in the cytoplasm, significant amounts are also measured in other subcellular compartments, such as the mitochondria (nearly 10%), nuclear matrix, peroxisomes, and the endoplasmic reticulum (ER) [38,39,40]. In these compartments, the GSH/GSSG ratio may differ notably from that in the cytoplasm according to their specific function. Additionally, many cells, including astrocytes, release GSH, as well as GSSG, using ATP-dependent multidrug resistance proteins (MRPs) from the ABCC family [41,42,43]. 

The importance of glutathione for cell function relies in its multiple roles (reviewed in [34,44]) that include the following: (i) control of the intracellular redox balance; (ii) detoxification of xenobiotics; (iii) antioxidant defense by free radical scavenging; (iv) maintenance of thiol protein status; (v) acting as a non-toxic reservoir of Cys [45]; (vi) modulation of DNA synthesis, microtubular-related processes or immune function; (vii) modulation of the activity of neurotransmitter receptors; (viii) control of NO homeostasis [46]; (ix) control of proliferation, apoptosis and necrosis; (x) acting as a reservoir for the glutamate neurotransmitter; and (xi) regulation of redox signaling by modulating protein activity through posttranslational modification (e.g., S-glutathionylation). Of special importance for the nervous system, the role of glutathione itself as a neurotransmitter has been postulated based on results obtained in several studies using, among others, radioactive ligands (e.g., [^3^H]-GSH) (reviewed in [47,48]). These studies showed concentration-dependent neuromodulatory actions of glutathione on glutamate receptors and even postulated the existence of glutathione receptors.

Several reports have also shown the essential role of the mitochondrial GSH pool in the defense against oxidative stress [49,50], for which these organelles rely on the import of cytoplasmic GSH and the GSR activity. Mitochondrial dysfunction is noticed in many pathologies that concur with associated redox stress, including neurodegenerative diseases (e.g., Parkinson’s and Alzheimer’s diseases) [51]. Patients suffering from these diseases, independently of their origin, often show nutritional deficiencies with the potential to impair mitochondrial function (e.g., coenzyme Q_10_ and niacin deficiencies) and antioxidant defense (e.g., selenium deficiency) [52]. Therefore, it is not surprising that ATP-dependent metabolic pathways such as GSH synthesis are affected by these conditions, especially in cells relying on the ETC for energy production (e.g., neurons). In fact, decreased GSH levels are commonly detected in these pathologies, although such reductions also occur during several physiological processes (e.g., aging) [51,53], thus stressing the importance of glutathione for cell/organ function. 

### 2.1. Glutathione Synthesis and Regulation

Glutathione synthesis requires three non-essential amino acids, Glu, Cys, and Gly and is carried out in two consecutive ATP-dependent reactions catalyzed by γ-glutamylcysteine ligase (GCL; EC 6.3.2.2) and glutathione synthetase (GSS; EC 6.3.2.3) (Figure 1). The process depends on Cys availability and the activity of the rate-limiting heterodimeric GCL enzyme. GCL establishes a peptide bond between the γ-carboxyl group of Glu and the amino group of Cys, leading to the γ-Glu-Cys dipeptide. In a second step, homodimeric GSS catalyzes the formation of another peptide bond between the carboxyl group of Cys in the γ-dipeptide and the Gly amino group, producing the reduced GSH form. Although the catalytic GCLC subunit is active *per se*, binding of the inactive modifier GCLM subunit decreases its affinity for Glu placing it within physiological levels of the amino acid (5–10 mM in the cytoplasm [54]). As reviewed by Dalton et al. [55], the importance of this effect could diverge among organisms given the differences in the kinetic behavior of GCLCs of various origins. Thus, e.g., the GCLC K_m_^Glu^ values and intracellular Glu levels are higher in rats than in humans and, in contrast to rat enzymes, the V_max_ of human GCL is 5-fold higher than that of GCLC. Moreover, GCLM binding also diminishes GCLC susceptibility to feedback inhibition by GSH [56,57,58]. Conversely, the oxidized GSSG form results from the use of GSH in certain processes, such as thiol-disulfide exchange. To keep the adequate GSH/GSSG ratio, GSSG can be reduced back into GSH by the action of the FAD^+^-dependent enzyme GSR (EC 1.8.1.7) using NADPH, both in the cytoplasm and the mitochondria.

Regulation of GSH synthesis occurs at transcriptional (which is cell-specific [59]), posttranscriptional, and posttranslational levels. Under oxidative stress, several redox-sensitive transcription factors (e.g., Nrf2) are known to induce the expression of *GCL* subunits and/or *GSS* [33], and stabilization/destabilization of *GCL* mRNAs has been also described (e.g., by 4-hydroxynonenal (4-HNE)). In astrocytes, thyroid hormone differentially regulates the expression of *GCL* subunits, inducing *GCLM*, in turn favoring the GCL hetero-oligomer [60]. Posttranslational modification by phosphorylation or caspase cleavage also modulates GCL activity (reviewed in [44]); e.g., increased astrocytic cytoplasmic Ca^2+^ levels inhibit GCL by means of CMK phosphorylation [61]. Moreover, peroxynitrite inhibits gerbil GSS in vitro [62], which is an important observation since its levels, as well as those of 4-HNE [63], are increased in several neurodegenerative diseases (e.g., Alzheimer, Parkinson, and amyotrophic lateral sclerosis (ALS)) [64].

### 2.2. Availability and Uptake of Amino Acids for Glutathione Synthesis

As mentioned previously, three non-essential amino acids are required for glutathione synthesis. Hence, their availability not only depends on their absorption from the diet, as there are endogenous metabolic pathways that allow their production from other sources or their recycling through protein degradation (Figure 2). These metabolic pathways may exhibit differences in activity among cell types, mainly due to diverse degrees of expression of the corresponding genes, thus reducing their potential utility and making the amino acid “essential” in that precise cell type. 

#### 2.2.1. Cysteine Uptake and Synthesis

Cys is mainly obtained from the diet, but cells with an active reverse transsulfuration pathway can also synthesize the amino acid from methionine [65] (Figure 2). Both, in cells and extracellularly, Cys can be found in its reduced SH-free and oxidized disulfide-linked (cystine, Cis) forms. However, the relative abundance of each form depends on the redox potential of the environment. Extracellularly, Cis concentration (40–50 μM) exceeds that of Cys (8–10 μM), due to rapid autoxidation of the latter [66]. Importantly, this autoxidation also takes place under standard cell culture conditions (half-life 30–60 min) [67,68], where the media contain undetectable amounts of Cys. In contrast, the reducing redox potential of the cytoplasm favors the Cys form that occurs at low micromolar concentrations [16], 10–100 fold lower than those of GSH. Measurements in samples of rat carotid artery and jugular vein evidenced the existence of Cis uptake into the CNS, although only Cys and GSH were found in cerebrospinal fluid (CSF) [69].

Cells require appropriate transport systems for Cys or Cis import [70], the differential expression of which determines the form of the amino acid that will be taken up. Cis uptake can take place in exchange for Glu through the Na^+^-independent and Cl^−^-dependent antiporter X_C_^−^ [71,72]. This system is composed of two disulfide-linked subunits, namely, the light xCT subunit (SLC7A11; note that the SLC nomenclature for amino acid transporter genes has been used for some proteins to unify the text), responsible for substrate specificity and the heavy 4F2hc subunit (also named rBAT or SLC3A2), common to all members of this transporter family. Induction of the X_C_^−^ system has been reported in response to oxygen and oxidative stress [73,74]. Importantly, the main role of X_C_^−^ is in redox buffering the Cys/Cis ratio, a function that can be modulated by competitive inhibition with Glu and homocysteic acid [75], as well as by its oxidative stress-induced expression [76,77,78,79]. 

Additionally, Cys can also be imported by cells using excitatory amino acid transporters (EAATs, also named X_AG_^−^ system). These highly conserved Na^+^-dependent Glu transporters are homomultimers, mainly trimers, in which each transmembrane subunit works independently [80]. There are five types that cotransport three Na^+^ ions together with one proton and one Glu while moving one K^+^ ion in the opposite direction [80]. Among them, human EAAT3 (also known as EAAC1 and SLC1A1) transports Cys at a similar rate than Glu upon expression in *Xenopus laevis* oocytes and with higher affinity for Cys (193 μM) than human EAAT1 (1830 μM, also named GLAST or SLC1A3) and EAAT2 (967 μM, also known as GLT-1 or SLC1A2) [81]. These three EAATs are redox-regulated by modification of Cys residues in their structure, oxidative stress inhibiting Glu uptake [82]. Moreover, EAAT3 is upregulated by SGK1 and PDK1, but downregulated by the δ-opioid receptor, GTRAP3-18, and the PI3K inhibitor worthmannin (reviewed in [83,84]), while its translocation to the plasma membrane is controlled by activation of PKC or PDGF receptors [85]. Nevertheless, regulatory effects of PKC on EAAT3 depend on the kinase isoform activated [86] and, additionally, may result in opposite behaviors regarding localization and activities of distinct EAATs (e.g., EAAT3 increase vs. EAAT2 decrease during PMA stimulation) [87,88]. Certain protein–protein interactions also regulate EAAT3 translocation, and thus interactions with reticulon protein RTN2B enhance its exit from the ER to the cell surface, binding to Rab1 supports ER-Golgi trafficking, and interactions with GTRAP3-18 maintain the protein at the ER (reviewed in [83,84]). 

Furthermore, neutral amino acid ASCT transporters that are Na^+^-dependent exchangers of small neutral amino acids also allow Cys uptake. ASCT1 (also named SCL1A4) is able to import Cys and presents developmentally regulated expression in several mouse cell types [89]. On the other hand, Cys is a competitive inhibitor of the ASCT2 system (also named SCL1A5) preferentially expressed in mouse peripheral tissues [90].

Endogenous Cys synthesis through reverse transsulfuration requires conversion of the essential amino acid methionine into homocysteine, an intermediate in the methionine cycle, that is also the link with the folate cycle (reviewed in [65]) (Figure 2). All these pathways are regulated by oxidative stress at different steps. Reverse transsulfuration is a cytoplasmic pathway that involves two homotetrameric vitamin B_6_-dependent enzymes, cystathionine β-synthase (CBS; EC 4.2.1.22) and cysthathionine γ-lyase (CTH; EC 4.4.1.1). CBS uses Ser and homocysteine to produce cystathionine, which, in turn, is utilized by CTH to render Cys, α-ketobutyrate, and ammonia. Additionally, both enzymes are able to produce the gasotransmitter H_2_S by using Cys as a substrate [91,92]. The existence of an active reverse transsulfuration pathway is a subject of controversy in different tissues and cell types, where very low expression levels have been detected for either enzyme of both. Hence, the Cys supply for glutathione synthesis using this pathway may show large variations depending on the tissue/cell type, reaching up to nearly 50% in the liver [93,94,95]. 

#### 2.2.2. Glutamate Uptake, Exchange, and Synthesis

Extracellular Glu concentrations of 7 μg/mL (46 μM) were reported in human plasma already in 1954 [96], whereas changes from 10 μM to 1 mM were found during depolarization in the synaptic cleft [97]. In the cytoplasm, the range established is 5–10 mM depending on the cell type [98,99]. Cells use Glu for protein and glutathione synthesis, but also as an energy substrate. In the brain, Glu has additional roles as an excitatory neurotransmitter released by glutamatergic neurons, which needs to be rapidly removed from the synapse to avoid neurotoxicity [82], and as a precursor in the synthesis of γ-aminobutyric acid (GABA), an inhibitory neurotransmitter [100]. Studies employing a variety of inhibitors have shown that, if needed, glutamatergic neurons can divert Glu from glutathione synthesis to neurotransmission [101]. Several Glu transporters/exchangers have been identified, including the already mentioned X_C_^−^ antiporter and different members of the EAAT family. Their abundance and even their cellular and subcellular localization may differ between cell types according to their specific roles, as described in the brain [102]. 

Endogenously, Glu can be produced by two main reactions involving the mitochondrial enzymes glutamate dehydrogenase (GDH; EC 1.1.1.3) and glutaminase (GLS; EC 3.5.1.2), but also from glucose, although the map of reactions and pathways that may lead to Glu is much wider, as reviewed by Yelamanchi et al. [103] (Figure 2). In fact, three enzymes are able to catalyze the reversible reaction allowing Glu synthesis from α-ketoglutarate, an intermediate of the tricarboxylic acid cycle (TCA). Those are GDHs [104], glutamic-pyruvic (GPTs), and glutamic-oxaloacetic transaminases (GOTs) [103]; these transaminases are feedback inhibited by Glu [103]. There are two GDH isoenzymes, but only the one codified by the *GLUD2* gene is specific to the nervous system [104]. Data from the Human Protein Atlas show that *GPT* expression is very low in the brain, but the two GOT proteins can be found at high levels in certain brain cells. This fact highlights the need to keep in mind that transcript levels do not always correlate with protein levels.

#### 2.2.3. Glycine Uptake and Synthesis

Gly is used for glutathione, protein, purine, heme, and creatine synthesis. Moreover, in the CNS it also acts as an inhibitory neurotransmitter and, together with Glu, as a coactivator of N-methyl-D-aspartate receptors. Additionally, Gly is involved in nerve energy metabolism. Its levels have been reported for human plasma (200–300 μM) [105,106,107] and pediatric CSF (9–23 μM) [108,109]. These concentrations should be controlled because their elevation may result in neurological disorders, such as those resulting from inborn errors of its catabolism [110]. Import of Gly into cells depends on several transporters of the ASC family, such as GLYT, B^0^AT, and VIAAT systems [70], which, according to the Human Protein Atlas, show differential transcript levels between cell types. 

Cells also synthesize Gly from Ser, Thr, hydroxyproline, and the macronutrient choline upon their uptake, although Ser can be alternatively obtained from glucose or Glu (reviewed in [111]). Human plasma levels reported for the precursor amino acids Ser (100 μM) and Thr (75 μM) are lower than those of Gly [105,106,107], whereas those for Ser concentrations in CSF seem larger (20–105 μM) ([108] and references therein). Synthesis of Gly from Ser can be catalyzed by cytoplasmic and mitochondrial serine hydroxymethyltransferases (SHMT and SHMT2, respectively, EC 2.1.2.1), which depend on vitamin B_6_ and tetrahydrofolate; it seems that only SHMT2 is expressed in most tissues (Figure 2). On the other hand, degradation of Thr by threonine dehydrogenase (TDH; EC 1.1.1.103) also allows Gly production, but its quantitative contribution seems to be low and to differ among species and their developmental stage. In fact, estimations in humans suggest that TDH may provide no more than 10% of the Gly pool [112]. Oxidation of the macronutrient choline is another source of Gly in which betaine, dimethylglycine, and sarcosine occur as intermediate metabolites produced by cytoplasmic and mitochondrial enzymes involved in the methionine and folate cycles. 

### 2.3. The γ-Glutamyl Cycle

As already mentioned, both GSH and GSSG can be exported from cells using MRPs (also known as ABCC system) (Figure 1). Once in the extracellular space, the γ-Glu-Cys bond, which is resistant to intracellular cleavage, is hydrolyzed in an ATP-dependent reaction catalyzed at the external surface of certain cells by γ-glutamyltranspeptidase (GGT; EC 2.3.2.2) [33,113]. Expression of *GGT* is induced by oxidative stress [114,115]. This step is part of the γ-glutamyl cycle described by A. Meister in the 1970s [33] that allows the export of GSH and recovery of the constituent amino acids as γ-Glu-X dipeptides (where X is any amino acid), as well as Gly plus Cys upon cleavage of the remaining Cys-Gly dipeptide by dipeptidase (DPEP; EC 3.4.13.19) at the membrane. Intracellular metabolism of γ-Glu-X dipeptides renders X and 5-oxoproline, which can be converted back to Glu in a reaction catalyzed by ATP-dependent oxoprolinase, hence allowing resynthesis of GSH. The high-energy cost (4 ATP molecules/turn) of the γ-glutamyl cycle suggests that its role in amino acid transport across the membrane may be minor in tissues such as the brain. Moreover, the content of the enzymes involved differs between brain areas [116], e.g., they are abundant in rabbit choroid plexus, suggesting a role in amino acid transport between the blood and CSF. Additionally, although the predominant role of GGT seems to sustain the availability of GSH precursors [117], it may contribute to extracellular detoxification [118], since the Cys-Gly dipeptide reacts faster than GSH with certain compounds [119].

## 3. General Facts about Astrocytes and Neurons and Their Energy Production

Among the variety of cell types encountered in the brain, neurons and astrocytes and their intercommunication have been the focus of many studies using diverse experimental approaches. Neurons, astrocytes, as well as oligodendrocytes, originate from radial glial cells (RGCs) that exhibit epithelial and glial features and which, in turn, derive from neuroepithelial cells of the embryonic neural tube [120,121]. Processes of RGCs also guide neurons from their ventricular origin to their final position which depends on their function. Neurons are considered “post-mitotic cells” in charge of obtaining and transmitting information throughout the body, as well as the processing of such information in the brain. For this purpose, they use electrical impulses and neurotransmitters that are released into the synapses, which can be excitatory (e.g., using glutamate) or inhibitory (e.g., using GABA or Gly); the former type seems to be the most abundant [54]. Neurons are in close contact, not only with other neurons but also with other cell types of the nervous system, which help them in their function and/or maintain brain homeostasis, as is the case of astrocytes. 

Neurons are unable to store glucose as glycogen due to the hyperphosphorylation and ubiquitin-dependent degradation of glycogen synthase (reviewed in [122]). Additionally, they prefer to divert their glucose consumption from glycolysis to the pentose phosphate pathway by constant degradation of 6-phosphofructo-2-kinase/fructose-2,6-biphosphatase 3 (PFKB3) [123] and hence the use of this fuel in an antioxidant pathway rather than in an energy-producing route. This is an important fact to sustain the NADPH levels required by GSR for GSSG reduction. To obtain energy, neurons seem to rely on pyruvate obtained from lactate to feed the mitochondrial TCA and, in turn, the ETC [122]. The high amount of energy they consume occurs mostly at the synapse [124,125], to preserve ionic gradients and for recycling (including uptake) of neurotransmitters, imposing the use of large amounts of oxygen [11,12]. This high oxidative rate increases the probability of enhancing ROS levels, which together with the limited antioxidant capacity of neurons compared, for instance, to hepatocytes (e.g., much lower expression of catalase or SOD1 and 50% less glutathione content) ([126], https://www.proteinatlas.org/ (accessed on 14 November 2022)) makes them highly susceptible to oxidative stress derived damage. Another characteristic that contributes to this high susceptibility is the elevated content of polyunsaturated lipids in neuronal membranes, which are especially sensitive to increased ROS content. Moreover, lipid peroxidation products, such as 4-HNE or acrolein, are more stable than ROS, allowing the spreading of their deleterious neurotoxic effects far away from their site of origin. Nevertheless, the activity of several neuronal antioxidant enzymatic systems can be induced by factors with neurotrophic activity such as PDGF [127], and differences among neuronal types regarding their degree of susceptibility to oxidative stress have been reported [128]. Besides the important expenditure of energy used for synaptic transmission [125,129,130] and restoration of membrane potentials after depolarization [124], neurons also invest a significant amount of energy in vesicle recycling, the synthesis of neurotransmitters and axoplasmic transport ([26] and references therein). 

Astrocytes became the dominant glial population phylogenetically as late as in birds and mammals [131]. Their functions are more diverse than those of neurons (reviewed by [132]) including (i) synapse formation and modulation (e.g., taking up the Glu neurotransmitter); (ii) brain metabolism (including that of GSH); (iii) defense against oxidative stress; and (iv) homeostasis of essential metals and water. Astrocytes can be found in resting or activated (reactive) states. Glial activation is induced by a variety of stimuli, including homocysteine and H_2_S (e.g., controlling vasodilation), the levels of which increase in several neurological and neurodegenerative diseases [133]. Reactive astrocytes release inflammatory cytokines and NO, which contribute to neuronal damage by precluding axon regeneration, increasing lipid peroxidation, and impairing mitochondrial functions, among others [134]. 

Astrocytes can be excited by changes in Ca^2+^ levels induced by diverse stimuli (e.g., neurotransmitters or mechanostimulation) to control synaptic signaling and neurotransmitter levels, but this process is bidirectional and astrocytes also release gliotransmitters that, in turn, modulate neuronal behavior (e.g., neuronal excitability) [135]. This constitutes the basis of the “tripartite synapse”, in which reciprocal intercommunication between astrocytes and neurons is established [135,136,137,138]. During neuronal activation, astrocytes help maintain neurotransmitter stores by e.g. rapidly clearing the Glu released by neurons at the excitatory synapses for its conversion into Gln, which is released back to the extracellular space for neuronal reuptake and Glu resynthesis [139,140,141]. A direct consequence of this process is an increased intra-astrocytic Na^+^ concentration that mediates enhanced glucose uptake for its use in anaerobic glycolysis, the main pathway used for astrocytic ATP production [11,12]. As a result, lactate concentrations increase in regions of brain activity, and its transfer from astrocytes to neurons has been postulated in order to feed neuronal oxidative metabolism [142,143,144]. Nevertheless, glucose has been reported to better support neuronal function than lactate in cell culture [145,146], as well as to fuel certain neuronal networks [147,148]. In addition to glucose, it seems that a large amount of the Glu imported by astrocytes is used for energy production, a mechanism favored by the formation of protein complexes involving the EAAT2 transporter, hexokinase, and mitochondrial enzymes (reviewed in [97]). 

Recently, it has been shown that local astrocyte membrane depolarization can occur at peripheral processes at the synapses with inhibition of Glu uptake, in turn, contributing to enhancing neuronal activation [149]. Astrocytes also maintain brain homeostasis through their involvement in the “neurovascular unit” at the blood-brain barrier, which is composed of endothelial cells, pericytes, and neurons communicating through gap junctions and hemichannels [138,150]. This communication allows the control of trafficking (e.g., metabolites, ions, and waste products) and blood flow, for which astrocyte endfeet wrap around endothelial cells and pericytes in arterioles and veins. As the endfeet of astrocytes surround the brain capillaries, these cells are among the first to obtain substrates for ATP production, as well as amino acids from the blood, both required for GSH synthesis. However, for this same reason, they are exposed earlier to drugs or toxic compounds that cross the blood-brain barrier. Clearance of these substances requires in many instances GSH-dependent detoxification, and hence the presence of higher levels of this metabolite in astrocytes than in neurons as will be explained in Section 4. Moreover, astrocytes also release glutathione, which aids in the control of unwanted effects provoked by oxidative stress in neurons (putatively providing GSH precursors) and endothelial cells of the blood-brain barrier (keeping its stability) [151]. Importantly, the neuroglial signaling in adults differs from that occurring during development, as has been reported in response to Glu signaling [152]. 

## 4. Glutathione Levels and Synthesis in Brain: Neurons and Astrocytes

The interest in determining glutathione levels in the brain has increased since its altered homeostasis was described in neurodegenerative diseases [153] and during aging ([154,155]). These measurements have been carried out in vivo and in tissues of different mammals. In the latter, concentrations of both GSH and GSSG have been determined using diverse methods and samples that were obtained immediately after death or preserved frozen. This variety of procedures has been postulated as the underlying cause of differences in the glutathione levels encountered in phylogenetically close organisms such as monkeys and humans [156]. Additional measurements have been also carried out in CSF samples (445 vs. 327 nM for control and schizophrenia patients, respectively) [157] and in enriched brain cell cultures, which have been widely utilized in studies on glutathione synthesis and metabolism as will be explained in Section 4.2.

### 4.1. Glutathione Levels and Distribution in Brain

The concentration range determined for GSH in brains of rats, monkeys, and humans is 1–3 mM [156,158], significantly lower than that measured in the liver (5–10 mM), the organ where most studies on glutathione have been carried out [37,159]. In vivo studies, although scarce, have been carried out on human control volunteers using ^1^H-NMR, rendering estimates of 1.3 μmol/g of GSH in the occipital lobe [158]; these values decrease in many neurodegenerative diseases [157]. Additionally, the distribution of GSH is not homogeneous and differences between its content in diverse brain areas and cell types have been found [160], as well as discrepancies in the evolution of its concentrations during aging which e.g. decrease in rodents and increase in humans ([154,155] and references therein). Regarding GSSG, brain levels reaching approximately 1% of those of GSH have been reported in rodents and humans [156], a proportion that increases during aging [155]. Altogether, these data indicate that GSSG values in the brain are similar to those in other tissues, placing the normal GSH/GSSG ratios also around 100.

Attempts to localize GSH in tissue sections using histochemistry revealed staining of neuropils of the nervous system, whereas different staining levels were found between neuronal somata of the CNS (low signals) and the peripheral nervous system [161,162]. Anti-glutathione antibodies also detected high glutathione staining in rat astrocytes, whereas labeling of neurons was more variable in cell bodies with substantial staining in the perikaryon [160]. The lower abundance of GSH in neurons than in glial cells was also reported in studies analyzing its concentration during rat postnatal development in various brain structures [29]. However, analysis of brain GSH levels from different organisms showed no correlation with their neuron density and, in fact, they were similar [29]. Additionally, estimates of the intracellular and extracellular GSH concentrations have been made by comparing data obtained from isolated striatum (1.37 mM) and in vivo microdialysis perfusates (77.2 nM–2 μM) [163,164]. These studies showed intracellular accumulation not only of GSH but also of γ-dipeptides such as γ-Glu-Cys. 

Glutathione release was identified in superfusates of rat brain slices of different regions under basal conditions, as well as stimulation of this efflux in response to K^+^ depolarization and by decreasing Ca^2+^ levels [165]. Anoxia and aglycemia also increased glutathione efflux from rat hippocampal slices, an effect reverted by the reintroduction of oxygen and glucose [166]. Perfusion of ETC inhibitors or FeSO_4_ did not alter basal extracellular levels of GSH in microdialysates of rat striatum and substantia nigra, but discontinuation of the treatments induced a transient release [167]. Concomitantly, efflux of Cys was observed under these conditions, as well as an increase in Glu levels if complex I inhibitor 1-methyl-4-phenylpyridinium (MPP) was used in the perfusion step.

### 4.2. Glutathione Metabolism in Brain Cells

As already mentioned, most data on glutathione levels and metabolism come from studies carried out in cell cultures enriched separately in either neurons or astrocytes, but also upon their coculture in transwell systems. For this purpose, neurons and astrocytes obtained at different developmental stages were generally used, i.e., rats or mice embryos (days E16–E18) for neurons (eN) and newborn animals for astrocytes (nA). In some cases, differentiation of astrocytes was induced with dibutiryl cAMP (dBcAMP), whereas their growth may have been restrained in eN cultures by cytosine arabinoside addition for 24 h. Differences in the culture timespan can also be found, which may be up to 7 days for eN. In contrast, this period extended to 14–21 days for nA, with no changes in glutathione content reported [168]. These are only some examples of the most obvious disparities in experimental approaches found in the literature, which may also include important differences in glucose concentrations (5 mM vs. 25 mM) in the media that could impact nA glutathione content [168]. Nevertheless, these cultures have provided interesting data regarding glutathione metabolism, as will be described below. 

#### 4.2.1. Glutathione Levels in Brain Cells

Numerous attempts have been made to determine glutathione levels in primary cultures of neurons and astrocytes obtained from different organisms and/or brain areas. High GSH levels (2–10 mM), comparable to those found in hepatocytes, have been reported in astrocytes in culture [45,169,170]. Moreover, higher glutathione levels were consistently measured in nA compared to eN from rats and mice [30,72,171,172], a pattern also found in cultures of chicken eA and eN [31]. Total glutathione levels of 32.8 nmol/mg were obtained by Dringen et al. using enriched rat nA primary cultures [168], whereas measurements in enriched eN (E16) primary cultures were highly variable with a mean value nearly 45% lower than that found in nA [173]. These authors used estimates of cytosolic volume (4.1 μL/mg) obtained with 3-O-methylglucose to calculate an 8 mM glutathione concentration in nA [168]. Analogous calculations in eN will render a 5 mM glutathione concentration, which is slightly higher than previously reported [29]. In contrast, Langeveld et al. only found lower GSH levels in neurons versus astrocytes isolated from the rat brain cortex, while no such difference was detected in cells obtained from the striatum or mesencephalon [160]. Moreover, other authors reported no big differences in GSH levels using nA and eN obtained from the cortex of rats of slightly different ages (P2 for astrocytes and E18 for neurons) [69]. Recently, some studies have started to report total glutathione levels in human iPSC-derived astrocytes (10 nmol/10^6^ cells), which were higher than those of undifferentiated neuroepithelial stem cells [174], but no additional comparisons were provided.

Discrepancies in intracellular glutathione content have also been detected between undifferentiated and dBcAMP-differentiated mouse nA, the latter exhibiting higher levels [30]. The length of culture was also found to contribute to the diversity of results described in different studies. Thus, increasing the time of culture did not significantly affect intracellular glutathione nA levels [175], although this parameter decreased in eA and more severely in eN [72]. Both nA and eN glutathione levels are susceptible to glucose or amino acid deprivation, but the decrease induced by these treatments is observed earlier in eN than in nA [171]. The glutathione content of nA is increased by oxidative stress, but it remains unaltered by the addition of Ca^2+^ [176]. A slight decrease or no decrease in intracellular glutathione levels was also detected in nA cultured in the presence of a GGT inhibitor, independently of the presence of Cys or Cis in the growth media [177,178]. 

Regarding GSSG, chicken eA exhibit higher levels than eN [31], a pattern close to that reported in equivalent rat cells. In fact, other authors could only detect GSSG in rat nA [69]. Using their values for total glutathione and GSSG, it is possible to estimate a much higher GSH/GSSG ratio for eN than for eA or nA, and hence a potentially more oxidative intracellular environment in astrocytes than in neurons.

Studies carried out in cocultures of nA and eN showed an increase in eN glutathione content from 0 to 24 h of culture, while the enhancement in nA was only slight and transient [173]. Inhibition of GGT prevented this eN glutathione increase, which led authors to propose that the Cys-Gly dipeptide was the precursor used for GSH synthesis [173]. Moreover, a reduction in eN GSH levels was found upon exposure to NO-releasing nA, partially mimicking the effect caused by the addition of NO donors to eN [172]. 

Altogether, it is widely accepted that astrocytes are glutathione-rich cells compared to neurons. However, a review of the data available also suggests that these differences may not be general and that levels within neurons depend on each specific type and location. 

#### 4.2.2. Glutathione Synthesis and Recycling in Neurons and Astrocytes

Steady-state glutathione levels depend on the synthesis, recycling, use, and export of the metabolite in each cell type. Neurons and astrocytes express GCL, GSS, and GSR and, in turn, both cell types are able to synthesize GSH from their individual components and reduce GSSG [31,101,179] (Figure 3). Nevertheless, the ability of each cell type to respond to stress conditions may be different due to e.g. their diverse Nrf2 levels, which are key to the expression of these genes. According to the available data, there is developmental repression of Nrf2 in neurons after birth [180], and cortical neurons have a higher capacity for its degradation and much lower levels of this transcription factor than astrocytes [180,181]. RNAseq data showed upregulation of *GCL* subunits and *GSR* in astrocytes cocultured with neurons [179], thus suggesting neuronal signaling acting to stimulate astrocyte glutathione metabolism. However, as already mentioned in Section 3, there are differences in the pathways that each cell type prefers for the production of the ATP needed for glutathione synthesis; the coordination between antioxidant and ATP synthesizing pathways will not be detailed here as it has been the subject of previous reviews [182]. 

Neurons and astrocytes also differ in their capability to import and synthesize the GSH-constituent amino acids, as will be explained in the following subsections. Moreover, higher GCL activity and protein levels have been reported in chicken astrocytes compared to neurons [31], as well as impaired GCL activity and increased GSH utilization (GPX, GGT, GST activities) during brain aging [155]. Regarding GGT, the addition of its inhibitor acivicin to nA or eN cultures caused, respectively, a mild decrease (25%) [177] or no effect in their intracellular glutathione content [173], therefore suggesting either no expression or no role of neuronal GGT. Still, it should be kept in mind that glutathione levels in eN and nA, which have been used in many instances for follow-up of cell response, are influenced by culture conditions, such as glucose or amino acid deprivation [171] and length of culture [72]. 

#### 4.2.3. Sources and Uptake of Reduced and Oxidized Cysteine for Glutathione Synthesis in Neurons and Astrocytes

A number of studies indicate that neurons use preferentially Cys, while both the Cys and Cis forms of the amino acid can be utilized by astrocytes. Measurements of intracellular Cys levels showed a higher content in rat eA than in eN [74]. To decipher the origin of the Cys pool in each cell type, culture media with different amino acid compositions have been used. These studies mostly indicated that eA and nA are able to maintain their Cys levels in Cys-free media by using Cis [74]. Both Cys and Cis could be used by nA to recover the glutathione pool after incubation in a minimal medium [178] and N-acetylcysteine (NAC) can fully substitute Cys for this purpose [168,176]. Moreover, although Cis uptake and glutathione content could be induced by oxidative stress [176], no effect was detected in nA after decreasing Cis concentrations in the medium [183], despite the resulting displacement of the extracellular redox potential towards more reducing values. On the other hand, eN were reported to rely on Cys to sustain glutathione levels [74,173]. In fact, Cys refeeding enabled the recovery of the eN glutathione pool after amino acid deprivation, while the addition of Cis had no effect [171]. 

Nevertheless, other studies point towards the ability of both cell types to use either form of cysteine. In fact, eN degeneration and lysis were observed when the Cis content of the media was decreased or its uptake inhibited, the latter leading to a diminution of intracellular glutathione levels [183,184]. The presence of Cis in the medium was found to be essential for the time-dependent recovery of glutathione levels of rat brain embryonic cells in culture and to increase Cys levels in the medium [73,74]. Loss of intracellular glutathione with culture timespan in both eN and nA cultured separately was prevented by Cis supplementation and the levels even increased by the addition of Cys to the medium [72]. A comparison of cells grown in media supplemented with Cys or Cis showed uptake of both forms of the amino acid in eN and eA, although the rates measured were 10-fold higher in eA than in eN [72]. Moreover, in both cell types, the import rate for Cys was approximately 4.5-fold higher than that for Cis, while uptake rates for Cis in eA and Cys in eN were within a similar range [72]. Mixed cultures containing eN and glial cells also showed that most [^35^S]-Cis uptake occurs in the latter using a low-affinity transporter inhibited by D-Asp [183]. In light of these results, the larger increase in glutathione levels detected upon the addition of Cis to nA versus eN could be ascribed to the higher uptake rate exhibited by nA, rather than to lack of import into eN. 

The putative use of reverse transsulfuration as an alternative source of Cys for glutathione synthesis has also been explored, the variety of results obtained suggesting a scarce contribution of this pathway. Thus, Met could not substitute for Cys to increase GSH content in eN [185], an effect that could be explained by the lack or minute *CBS* expression detected in several neuronal types [186,187]. In nA, Met was unable to replace Cys for GSH production [176], higher rates of synthesis were obtained in the presence of Cis versus Met [188], and partial substitution was obtained by the addition of cystathionine to starved nA [168]. Additional reports described an active flux through reverse transsulfuration in astrocytes [93,95,189,190], which seemed to be low [191]. These last results match better with recent high-throughput proteomic analysis of mouse brain cells in which CBS and CTH were detected in astrocytes and neurons [192]. Thus, it seems that depending on the cell type, but also on the organism, a different degree of contribution of reverse transsulfuration to Cys pools could be observed. 

Differences in the expression of diverse transporters for Cys and Cis in each cell type have been also examined (Figure 4). In fact, expression of the xCT subunit of the X_C_^−^ antiporter, which has been detected in neurons and astrocytes of mouse and human cerebral cortex [193], seemed limiting for GSH synthesis in several brain cell types [66]. Moreover, uptake of Cis using this system was described for immature astrocytes [71] and, at very low levels, for immature neurons [72,183]. Upregulation of X_C_^−^ by decreased intracellular GSH concentrations was observed in nA [194]. Of note, in neurons, this system also imports cystathionine, the intermediate in reverse transsulfuration used by CTH for Cys synthesis, thus bypassing the potential lack of CBS already mentioned. 

Cys, considered the limiting substrate for neuronal GSH synthesis [173], is imported through EAAT in mature neurons [195,196]. Expression of *EAAT3* is induced by Nrf2, and hence influenced by the developmental repression of this transcription factor in neurons [180]. EAAT3 has been detected in C6 glioma cells and primary rat eN. Only a small percentage of this EAAT3 was located at the plasma membrane, and translocation was detected upon activation of PKC or PDGF receptors [85]. To stress the importance of this transporter in neuronal Cys uptake, silencing EAAT3 expression led to decreased uptake of this amino acid and diminished glutathione synthesis in cultures of cortical neurons [197], as well as abnormal brain development in adult null mice [198]. Additionally, ASC neutral amino acid transporters were also identified. ASCT2 seemed restricted to astrocytes ([83] and references therein), whereas ASCT1 was detected in astrocytes and embryonic neurons [89]. ASCT1 levels were maintained high in astrocytes during development, whereas their decrease was described in late embryonic and neonatal neurons [89]. Importantly, in rat brain embryonic cells in culture, the uptake rate of Cys by ASCT1 is much larger than for Cis using X_C_^−^ [74]. 

Altogether, the data summarized above suggest that neurons depend mostly on Cys for glutathione synthesis, while astrocytes are able to obtain the amino acid in several forms. These differences seem to rely on the diverse transport system expressed in each cell type and/or their levels and/or import rates.

#### 4.2.4. Sources and Uptake of Glutamate for Glutathione Synthesis in Neurons and Astrocytes

As explained in Section 2.2.2, Glu is involved in a large number of pathways that allow its synthesis from a variety of sources and its use in other pathways besides glutathione synthesis. To decipher Glu contribution to GSH production, experiments modifying the amino acid composition or using isotope labeling have been carried out. Results derived from changes in the amino acid composition of the media suggested that Glu is the limiting factor for nA glutathione synthesis, while eN prefers Gln [171]. However, the addition of [^15^N]-Glu to the media showed that nA favors its use for transamination into [^15^N]-Asp and conversion into [^15^N]-Gln and [^15^N]-Ala, instead of glutathione synthesis [169]. Moreover, this same study also revealed that cosupplementation with [^15^N]-Glu and Cys increased nA glutathione [^15^N]-labeling and concentration, thus suggesting that the Glu partition between pathways may depend on the availability of Cys. In fact, the sole addition of Cys to the nA medium resulted in a reduction in the intracellular Glu pool that was abolished by buthionine sulfoximine (BSO)[169], which inhibits the first step in GSH synthesis, supporting the notion that this decrease is due to consumption for GSH production. Higher levels of both Glu and Asp were detected in eN than in nA under conditions where no difference in their GSH content was observed, while Cys could only be identified in nA [69], a fact that could be indicative of the limiting role of Cys levels for eN GSH synthesis. 

Neurons and astrocytes of the CNS express GLS and hence are able to produce Glu from Gln [199]. Differences in GLS activity among neurons from different brain areas have been observed, as well as much higher activity in mouse neurons than in astrocytes in culture [200], a pattern opposite to that described in chicken eN versus eA [31]. In this reaction, neurotoxic ammonia is also generated and transferred from neurons to astrocytes for its efficient removal, a role performed according to the Glu/Gln cycle [97,100]. Additionally, only one GDH isoenzyme codified by the *GLUD2* gene that is able to use α-ketoglutarate for Glu synthesis is specific to the nervous system and can be detected by immunohistochemistry in astrocytes [104]. More recently, other authors also suggested the existence of lower expression and activity of GDH in neurons than in astrocytes [54]. Alternative routes to obtain Glu have also been explored and such studies showed that Asn, Asp, Pro and ornithine can be used as sources of the amino acid in starved nA [168]. Another way of producing Glu is through the use of glucose, a mechanism that seems precluded in neurons due to a lack of pyruvate carboxylase expression ([100] and references therein). However, the validity of this statement is put into question by the detection of the corresponding mRNA and protein in neurons and astrocytes reported in the Human Protein Atlas; in fact, according to this resource, protein levels in neurons are higher than in astrocytes. 

Among Glu transporters, several members of the EAAT family are expressed in astrocytes and/or neurons (Figure 4). According to the immunostaining of adult rat brain sections, EAAT3 is detected in pre- and post-synaptic structures of certain neuronal types, EAAT2 is found only in astrocytes, and EAAT1 is found in both neurons and astrocytes [102]. However, additional studies also identified EAAT4 and EAAT5 (also named SLC1A6 and SLC1A7, respectively) in Purkinje and retinal neurons, respectively [82], and EAAT3 diffusely in cell bodies and processes of astrocytes [102]. Data from the Human Protein Atlas also confirm differences in EAATs’ expression according to the cell type, as well as higher levels of Glu transporter systems in astrocytes than in neurons. 

System A members *SLC38A1* and *SLC38A2* for Gln transport are also expressed in human neurons and astrocytes and, according to the Human Protein Atlas, their levels are nearly 2-fold higher in both excitatory and inhibitory neurons than in astrocytes (Figure 4). Nevertheless, additional members of this family, as well as system N members, are also detected (although at lower expression levels). Differences in expression levels together with data on glutathione synthesis obtained from eN and nA in culture using media combining Cis or Cys with Glu or Gln [171] suggest that neurons prefer Gln over Glu, while astrocytes show the opposite behavior. 

#### 4.2.5. Acquisition of Glycine for Glutathione Synthesis in Neurons and Astrocytes

Current data of the Human Protein Atlas show similar transcript levels of several Gly transporters in neurons (B^0^AT system *SLC6A17*) and astrocytes (ASC system *SLC1A4*), although different systems are preferred in each cell type (Figure 4). Moreover, neurons also express significant levels of *SLC1A4*, while astrocytes exhibit notable expression of the ASC system *SLC7A10* (also named ASC-1) that seems to prefer Arg transport in these cells [70]. Additionally, inhibitory neurons express the VIAAT system SLC32A1 for Gly export at nerve terminals [70], GLYT1 (SLC6A9) is expressed in astrocytes (reviewed in [132]) and GLYT2 (SLC6A5) in glycinergic neurons [201]. The mRNA expression levels, however, do not correlate with protein levels reported in the Human Protein Atlas, the latter being highly dependent on the quality and affinity of the antibodies.

Alternatives to Gly import seem to rely on the use of Ser since the potential contribution of hydroxyproline conversion to the Gly pool appears poorly explored. In fact, Gly replacement by Ser was proven in starved nA [168]. Additionally, the utilization of choline oxidation in brain cells may be also limited by the low expression of the enzymes involved, e.g., betaine homocysteine methyltransferase (BHMT; EC 2.1.1.5) [65,111]. 

#### 4.2.6. Dipeptides as Precursors for Glutathione Synthesis

Astrocytes in culture are also able to synthesize GSH from a variety of dipeptides [202]. Starved nA can use Cys-Gly to replace Cys and Gly and γ-Glu-Cys as a substitute for Glu [168]. Nonetheless, the effect of these dipeptides on glutathione synthesis was only limited compared to the single amino acids [168,176]. Other studies also reported the ability of neurons to use γ-Glu-Cys and Cys-Gly, derived from GGT cleavage of GSH, for their own GSH synthesis [173]. In fact, glutathione content in eN was increased when Cys-Gly or γ-Glu-Cys were added to the culture media, although the concentrations required to reach the maximum levels were higher for the γ-dipeptide [173]. Uptake of the Cys-Gly dipeptide in astrocytes was ascribed to the use of the peptide transporter PepT2, while the same authors suggested that the dipeptide must be cleaved by an ectopeptidase for the import of the constituent amino acids into neurons [203]. Of note, very low dipeptidase transcript levels are reported for neurons and astrocytes in the Human Protein Atlas, although DPEP2 protein is found in glial cells. Further support for the use of dipeptides comes from the existence of similar GSS and GSR activities in chicken eA and eN [31], which confer these cells the putative ability to use at least the γ-Glu-Cys dipeptide for GSH synthesis, as well as to reduce GSSG. However, treatment with buthionine sulfoximine (BSO), a well-known inhibitor of GCL, surprisingly blocks the glutathione increase induced by the dipeptides in eN [173], suggesting that extracellular γ-Glu-Cys may not be used directly by neuronal GSS. Altogether, provided that the dipeptides imported in both nA and eN can be used for glutathione synthesis, this may represent a substantial sparing of energy.

#### 4.2.7. Glutathione Utilization in Detoxification

Some detoxification processes utilize GSH in enzymatic and non-enzymatic reactions, the former involving, for instance, glutathione peroxidases (GPXs) and glutathione S-transferases (GSTs). Among them, the elimination of hydrogen peroxide by GPXs and formaldehyde (with higher concentrations in the brain vs. blood) shows similar rates in both neuron and astrocyte cultures ([132] and references therein). Expression of GPX1 and GPX3 has been described in astrocytes in culture, while GPX4 is induced during brain injury. Moreover, data in the Human Protein Atlas show low expression of *GPX1* in astrocytes and neurons (excitatory and inhibitory), while detection of the protein is reported in neurons and neuropils of the cortex, Purkinje cells, and glial cells of the basal ganglion and hippocampus. Nevertheless, hydrogen peroxide clearance decreases with aging as measured in astrocytic cultures of mice, correlating with lower GSR activity and increased GSH export from cells of old mice [132].

Importantly, some studies have found that γ-Glu-Cys could substitute GSH for certain functions. Precisely, the roles of GSH as an antioxidant and in the prevention of neuronal death of eN and in vivo can be assumed by γ-Glu-Cys acting as GPX1 cofactor, whereas this γ-dipeptide is less efficient than GSH regarding protein modification [204]. In fact, both reduced and oxidized forms of this γ-dipeptide have been detected in tissues, including the brain [204] and, as mentioned previously, eN are able to import γ-Glu-Cys from the media [173]. 

#### 4.2.8. Export of Glutathione

The reported expression of glutathione transporters in the brain ([41], https://www.proteinatlas.org/ (accessed on 14 November 2022)) suggests that it can be exported to the extracellular milieu. These transporters belong to the MRP family (ABCC system) and show different abilities for the transport of either GSH or GSSG, as well as of GSH-conjugates. Thus, ABCC4 and ABCC1 (also known as MRP4 and MRP1, respectively) are able to export GSH, while only the latter excretes GSSG from astrocytes [43,205] and most GSH-conjugates in mice brains [206]. According to the Human Protein Atlas, both transporters are expressed in astrocytes and neurons (excitatory and inhibitory). *ABCC4* expression is nearly 3-fold higher in astrocytes than in neurons and the opposite is observed for *ABCC1* transcript levels. Nevertheless, protein detection is only reported in brain endothelial cells according to this database, whereas stimulation of glutathione efflux by formaldehyde has been shown in neurons in culture [207]. 

Estimations of kinetic parameters for the GSH carrier have been obtained in nA [175]. Moreover, the rate of efflux was shown to be temperature-dependent, sensitive to thiol-reactive compounds (e.g., mercurials), and to show a biphasic dependence on the intracellular GSH concentration [175]. GSH release from nA was inhibited by compounds blocking the ABCC1 carrier [208], but also by cations and gap junction inhibitors with a profile that suggested a putative role of hemichannels in this process [176]. Additional measurements carried out in primary cultures of other brain cells only showed a certain GSH efflux in oligodendroglial cells [208], and more recent studies using iPSC-derived astrocytes also found increased extracellular levels of GSH compared to undifferentiated neuroepithelial stem cell cultures [174], hence reinforcing the idea of astrocytes (nA or iPSC-derived) as suppliers/releasers of glutathione. 

## 5. Astrocytes as Suppliers of Glutathione Components: The Hypothesis

The concept of astrocytes as providers of precursors for neuronal GSH synthesis has important implications, both in physiology and pathophysiology. Notably, the role of astrocytes in neurodegeneration is being increasingly recognized, and various neurodegenerative diseases are astrocytopathies. Among them, Alexander disease entails particular interest because it originates in astrocytes [209]. In this disease, mutations in an astrocyte protein, glial fibrillary acidic protein (GFAP) produce a derangement of astrocytic functions, which, by lack of support or toxic effects of sick astrocytes, compromise the homeostasis of other brain cell types, leading to neurodegeneration [210,211]. According to the available studies on Alexander disease, one of the astrocytic functions reported to be compromised is the provision of GSH, an antioxidant that is key for the control of oxidative modifications to which certain GFAP mutant isoforms are more susceptible [212,213]. Thus, Alexander disease provides an example of great interest in understanding glutathione metabolism for the potential prevention and/or reversion of neurodegeneration.

Studies carried out around the 1990s with enriched primary cultures using ^13^N-tracers, computer simulations, and histochemistry, together with the rapid decrease in glutathione content of eN as they were separated from cocultures with eA or nA, led several authors to propose the role of astrocytes in keeping neuronal glutathione levels [30,72,161,214]. This proposal was also supported by the finding of increased GSH levels in eN cocultured with nA [172,173], enhanced glutathione concentrations in the media once maximal intracellular levels in eN were achieved [173], as well as the existence of higher extracellular Cys levels in eA than in eN dishes after 12 h in Cys-free medium [72]. In the presence of Glu, Cis uptake by eA is inhibited and the extracellular Cys increase is precluded, hence suggesting that this Cys is intracellularly generated from the Cis taken up from the culture media [72]. 

Cocultures showed also the ability of astrocytes to support other cell types in their antioxidant defense against several ROS-inducing compounds (reviewed in [203]) which, according to the calculations performed with the available data [215], was even achieved at astrocyte/neuron ratios of 1:20. This collaboration between both cell types in coculture further extends to the upregulation of genes involved in glutathione synthesis through Nrf2 [179]. This effect was shown to depend on cortical astrocyte Glu receptors upon binding of soluble factors derived from activated hippocampal neurons, although whether this neuronal signaling induces astrocytic GSH efflux was not evaluated. Cocultures with NO-activated nA also reproduced effects such as decreased intracellular eN GSH levels and increased cell death induced by NO donors, which also caused a modest reduction in eN ATP content [172].

Examination of glutathione levels in independent cultures using stable isotope-labeled tracers or different culture media compositions has provided additional data to support this hypothesis. Cultures carried out in the presence or absence of Cis showed extracellular glutathione only in nA media, but not in that of eN [69]. The addition of acivicin to nA cultures to inhibit cleavage by GGT allowed the detection of higher extracellular glutathione levels, and the concentrations increased over time [173,175,177,208,216]. Evaluation of glutathione levels in nA cultures indicated their initial intracellular decrease correlating with [^15^N]-Glu uptake (used as a tracer) [169]. This diminution was followed by an increase that surpassed the initial intracellular content and enhanced extracellular glutathione concentrations [169]. Depletion of intracellular glutathione in rat nA cultures with diethylmaleate resulted in its increased synthesis, followed by [^15^N]-Glu labeling and a concomitant decrease in the levels of its three constituent amino acids [169]. Additionally, this tracer also showed a decrease in the extracellular glutathione levels that may derive from diethylmaleate-treated nA diverting Glu for its conversion to other amino acids. GSH was found in the media of nA grown in the absence or presence of Cis, but not in that of eN [69]. Moreover, media of nA grown with Cis also contained Cys and the Cys-GSH conjugate, whose concentrations increased in parallel to a decrease in Cis level [69]. In contrast, measurements of eN media, after coculture with nA and subsequent refeeding with or without Cis, showed no thiols or Cis and tiny amounts of Cys-GSH, respectively [69]. These data were interpreted as suggestive of non-enzymatic reaction of Cis and some GSH exported by eN. Despite the fact that altogether these data suggested that only nA were able to release GSH, there are other reports in which GSH release from these cells was only detected early in culture, or after ischemia, coinciding with cell death [170]. Moreover, expression of appropriate transporters for glutathione release has been also detected in neurons (https://www.proteinatlas.org/ (accessed on 14 November 2022)) 

Data obtained from the use of iPSC-derived cells are slowly appearing in the literature. Similar to nA, human iPSC-derived astrocytes in culture increased extracellular glutathione levels compared to their undifferentiated counterparts [174], thus suggesting their ability to export the antioxidant. Moreover, cultures of human neuronal precursor cells (LUHMES) were protected from cell death induced by proteasomal inhibitors in coculture with mouse or human iPSC-astrocytes [217]. The addition of the iPSC-astrocytic culture media induced intracellular GSH levels in proteasome-inhibited LUHMES, an effect also observed in cocultures with or without contact between both cell types [217]. This same study also found that proteasome-inhibited LUHMES enhanced their intracellular GSH and Cys levels with the addition of 1 mM GSH, while 1 mM Cys led to an increase in its own intracellular levels. However, the relevance of these results may be hampered by the use of thiol concentrations that are far from those encountered extracellularly [16], and the inability of GSH to enter cells unless cleaved by GGT, whose activity in neurons seemed to be very low [173].

## 6. Putative Drawbacks of the Hypothesis and Additional Possibilities to Achieve Appropriate Neuronal Glutathione Levels

This appealing hypothesis of astrocytes as suppliers of glutathione precursors to neurons would imply GSH synthesis to be carried out twice, first in astrocytes and later in neurons, as well as the export and hydrolysis of this antioxidant before the final import of the required amino acids into neurons. As explained in previous sections, several steps involved in this process use ATP, and hence impose a large energy expenditure [33,34]. This energy cost could be halved by avoiding the use of the γ-glutamyl cycle. Authors in the field assume that this is needed because neurons are unable to import Cis, the main form of cysteine in the extracellular environment, while this is the preferred form taken up by astrocytes. However, as mentioned in Section 4.2.3, expression of the transporters for Cys and Cis has been reported in both cell types [83,89,193,195,196] and shown in the Human Protein Atlas (Figure 4 and Figure 5). Moreover, looking at the uptake rates in culture, both cell types import Cys and Cis [72], although neurons show 10-fold lower rates than astrocytes for each form of the amino acid. Hence, the problem for neurons seems to be a slower uptake of Cis (Figure 5). In fact, some authors report their growth on Cis media, although the initial decline in glutathione levels induced by Cys elimination is not reverted [171]. Additionally, there is no reason to believe that the Cys-Gly dipeptide resulting from GGT hydrolysis and/or the Cys obtained from its subsequent cleavage by dipeptidase remain in their reduced state due to the oxidative redox potential of the extracellular space [16]. Thus, is there any other reason for astrocytic glutathione release? Are there other means to supply astrocytic glutathione to neurons?

First of all, many cells are able to export glutathione, but its release can also occur during cell death. However, appropriate controls for cell death have not always been reported in the literature concerning this hypothesis. Nevertheless, among those reporting on this parameter, some detected no glutathione release from nA until there is cell death caused by ischemia [170], while others measured 4–6.8 μM total glutathione concentrations in the media with less than 5% cell lysis in the presence of 10–100 μM acivicin [173,177]. Taking into account the high intracellular GSH levels found in astrocytes (8 mM), a substantial increase in the concentrations detected in the media could be expected if cell death above a certain threshold takes place. Using values reported for mean cell volumes in the literature, the release from just 5000 dead astrocytes into a 0.1 mL containing well can be estimated to make a mid-nanomolar contribution to the glutathione level. 

Second, according to the hypothesis, GSH and/or its hydrolyzed products, the Cys-Gly dipeptide and Cys, should remain in their reduced state to be used by neurons. This assumption presents the problem that the release occurs in a rather oxidizing extracellular environment, and hence oxidation of the Cys-containing metabolites may occur within a short timeframe and/or as diffusion towards neurons occurs. To counteract this drawback of the hypothesis, it has been proposed that the substantial GSH release from astrocytes could serve not just as the source of appropriate forms of its constituent amino acids to neighboring cells, but rather to induce a local change of redox potential in the extracellular space [16,218], in turn facilitating conversion of Cis into Cys for its import into neurons. This decrease in the Cis/Cys ratio would be further promoted by the higher astrocyte demand for Cis in order to support GSH synthesis for release. Thus, a local loop would be created in which the higher astrocytic Cis uptake would favor a more reducing extracellular redox potential that, in turn, would contribute to maintaining the reduced state of the released GSH, promoting the local decrease in the Cis/Cys ratio and therefore a higher availability of reduced Cys species for neuronal uptake. These Cys-containing building blocks for neurons, however, may or may not be of astrocytic origin. Additionally, the local pH decrease induced by astrocytic lactate release may also contribute to maintaining the Cys-containing molecules in their reduced state. Other possibilities in this same line include the reduction in extracellular Cis by the membrane-permeable gasotransmitter H_2_S. This effect was proven by the addition of the gasotransmitter donor NaHS to cell culture media and the resulting eN intracellular increase in Cys, γ-Glu-Cys, and GSH content and GCL activity [219]. At first, using H_2_S to obtain Cys seems less useful since this amino acid is used in most cases for the synthesis of the gasotransmitter by means of CBS and CTH or after its catabolism to mercaptopyruvate by means of 3-mercaptopyruvate sulfurtransferase [91]. However, CTH is also able to use reduced and oxidized forms of homocysteine for this purpose. High homocysteine levels have been found in patients with neurodegenerative diseases [220], where neurotoxic effects were ascribed to protein N-homocysteinylation [221], hence making its elimination an important issue. Nevertheless, earlier reports also proposed that reduction of Cis is the main role of plasma homocysteine [222], and therefore it could be also suggested that high extracellular levels of homocysteine could be helpful to supply Cys to neurons, a postulate that is in sharp contrast with the already mentioned homocysteine neurotoxicity. 

Third, all cell types of the CNS release extracellular vesicles (EVs), which act as messengers for cell communication by binding to certain surface receptors or delivering their content (proteins, different types of RNAs, lipids, etc.) into the target cells [223]. Neurons are known to deliver EVs that reach astrocytes e.g. to regulate Glu uptake [224]. Conversely, astrocytic EVs participate in neuromodulation and also in neuroprotection e.g. under oxidative stress or nutrient deficiency [223]. Therefore, an alternative way to supply glutathione to cells could be the shuttling of EVs charged with the antioxidant and/or the amino acids needed for its synthesis. In fact, glutathione (GSH and GSSG) and Cys have been found in some of the metabolomic analyses of EVs from other sources carried out to date [225,226]. Moreover, the presence of S-glutathionylated proteins in EVs has been also described [227], and while this modification could serve initially as a signal for protein secretion [228], it can also become a way to deliver GSH into the target cell. For this purpose, reduction of the glutathionylated protein in the cytoplasm of the receiving cell would render GSH plus the unmodified protein (e.g., thioredoxin and peroxiredoxin). Additionally, it could be also speculated that glutathione-binding proteins detected in EVs may also serve as putative carriers of glutathione. Such an aspect has not been analyzed in the astrocyte/neuronal context.

Finally, the existence of functional gap junctions between neurons and astrocytes has been reported in culture settings and in brain slices [229,230,231,232], and such connecting structures also exist between neurons and microglia [233]. These structures allow the exchange of compounds of limited sizes (up to 1–2 kDa) well above those of GSH (307.3 Da) and GSSG (612.6 Da). Additionally, tunneling nanotubes may be another option to exchange glutathione between both cell types. These structures are formed under stress both in astrocytes and neurons and are proposed to be a defense mechanism for damaged cells (reviewed in [234]). Therefore, both gap junctions and tunneling nanotubes may constitute a putative alternative for a rapid glutathione supply from nearby cells to neurons, limiting the energy costs derived from the use of the γ-glutamyl cycle and ulterior neuronal GSH resynthesis. The transwell setting used in many coculture studies precluded contact between neurons and astrocytes, thus avoiding the possibility of testing these alternative mechanisms. These mechanisms could allow the transfer of GSH or its precursors without the need for their release into the extracellular space.

## 7. Conclusions

The use of enriched cultures of neurons and astrocytes has been a useful tool for studying their behavior under a variety of culture conditions. However, many discrepancies in the utilization of one or another amino acid for glutathione synthesis may arise from the lack of homogeneity of the cell populations used and/or their isolation from different brain areas and/or at diverse developmental stages. Additionally, the collected data indicate that the comparison and generalization of conclusions obtained from cells derived from different organisms may be risky, due to disparities in their behavior, gene, and protein expression. Given that most of the genes and proteins involved are regulated by stress (including oxidative stress), maximum care should be taken to evaluate the initial conditions of the cells used. Just a slight sign of stress in the cultures may lead to quite different outcomes, including those due to cell death. Data from cocultures mostly rely on the use of cell types obtained before (E16-E18 for neurons) and after birth (astrocytes), and hence that hardly will coexist in vivo. Moreover, this is a timeframe where many hormonal changes take place to adapt from the maternal environment to independent life, leading to alterations in the expression pattern of many key metabolic genes. In fact, key transcription factors (e.g., Nrf2) controlling the expression of glutathione synthesis enzymes are developmentally regulated, an aspect that may need to be considered to reach meaningful conclusions when dealing with this type of cocultures. Finally, the available data show that the fact that astrocytes supply glutathione precursors to neurons is still a hypothesis that requires further assessment before it is accepted as a fact. Moreover, additional aspects need to be considered and explored, including the putative role of changes in extracellular redox potential induced by glutathione release and/or the existence of additional or complementary mechanisms to deliver glutathione to neurons.

## Figures and Tables

**Figure 1 ijms-24-08059-f001:**
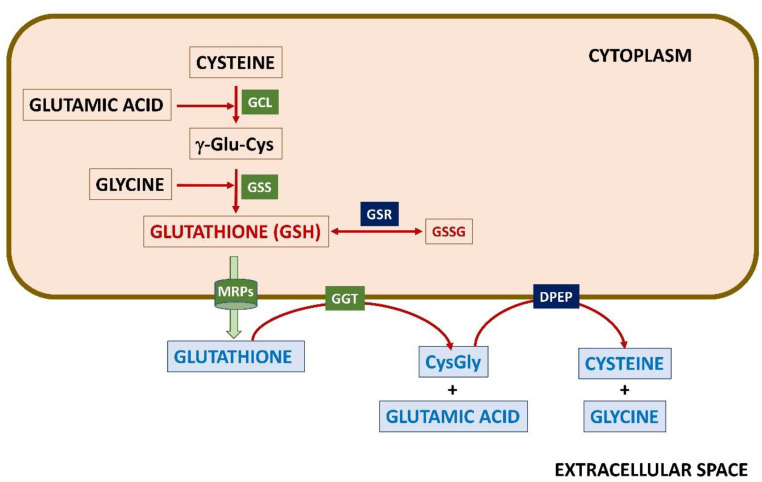
**Glutathione synthesis, export, and extracellular cleavage.** The figure shows the enzymes involved in the synthesis of the glutathione reduced form (GSH) and the reduction in its oxidized form (GSSG) in the cytoplasm. Transporters carrying out the efflux to the extracellular space are placed at the plasma membrane together with enzymes involved in extracellular GSH cleavage into its constituent amino acids. The proteins involved are γ-glutamylcysteine ligase (GCL), glutathione synthetase (GSS), glutathione reductase (GSR), γ-glutamyltranspeptidase (GGT), dipeptidase (DPEP), and multidrug resistance proteins (MRPs). ATP-consuming steps are indicated with a green background on the protein’s name. Extracellular metabolites are indicated in blue. Both GSH and GSSG are exported by cells.

**Figure 2 ijms-24-08059-f002:**
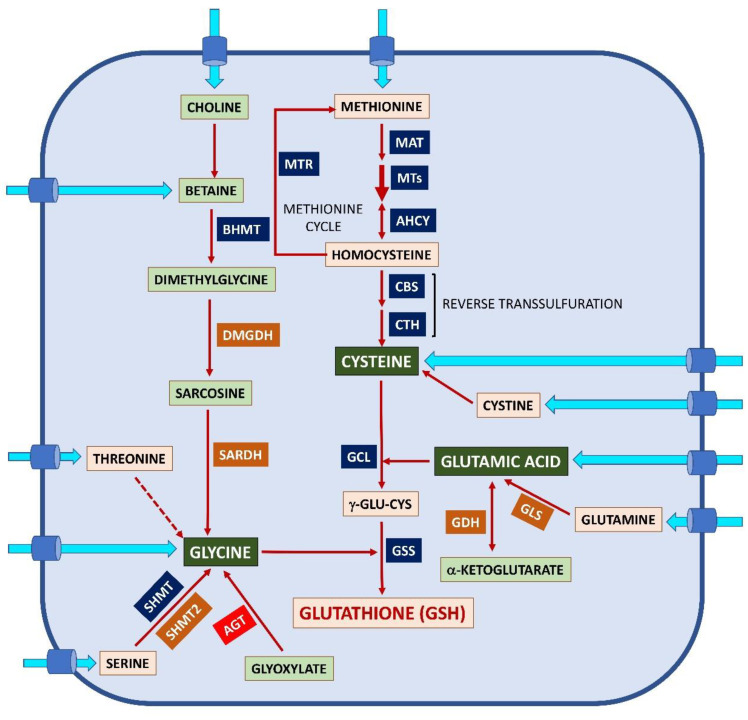
**Schematic representation of the main sources of amino acids for glutathione synthesis.** The figure shows the import of amino acids through transporters at the plasma membrane and the main pathways used for the cellular production of Cys, Glu, and Gly (green forest background) required for glutathione synthesis. For simplification, the contribution of protein degradation as an amino acid source is not shown and discontinuous lines indicate conversion through several steps in the route that are not detailed. Amino acids and peptides are depicted in a salmon background, whereas green was the color used for other metabolites. Cytoplasmic (dark blue), mitochondrial (orange), and peroxisomal (red) enzymes involved in these pathways are: AGT, alanine glyoxylate aminotransferase (EC 2.6.1.44); AHCY, adenosylhomocysteinase (EC 3.13.2.1); BHMT, betaine homocysteine methyltransferase (EC2.1.1.5); CBS, cystathionine β-synthase (EC 4.2.1.22); CTH, cystathionine γ-lyase (EC 4.4.1.1); DMGDH, dimethylglycine dehydrogenase (EC 1.5.8.4); GCL, γ-glutamylcysteine ligase (EC 6.3.2.2); GDH, glutamate dehydrogenase (EC 1.4.1.3); GLS, glutaminase (EC 3.5.1.2); GSS, glutathione synthetase (EC 6.3.2.3); MAT, methionine adenosyltransferase (EC 2.5.1.6); MTR; methionine synthase (EC 2.1.1.13); MTs, S-adenosylmethionine-dependent methyltransferases; SARDH, sarcosine dehydrogenase (EC 1.5.8.3); SHMT, serine hydroxymethyltransferase (EC 2.1.2.1); SHMT2, mitochondrial serine hydroxymethyltransferase (EC 2.1.2.1).

**Figure 3 ijms-24-08059-f003:**
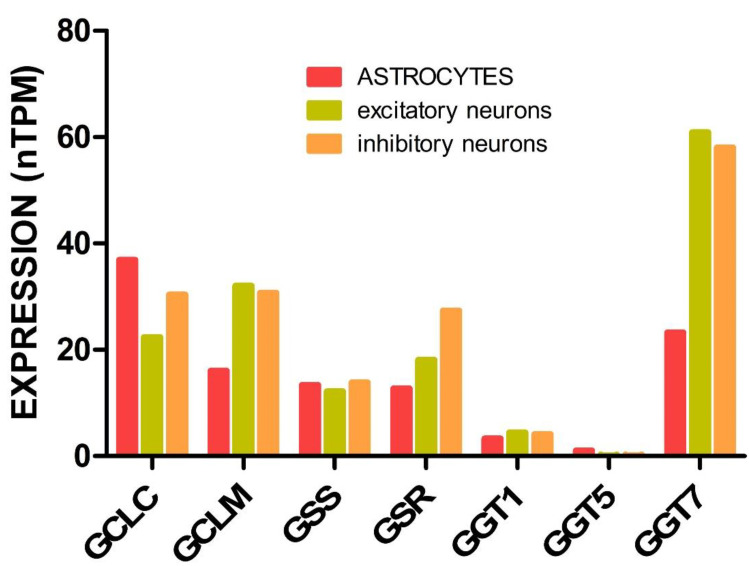
**Expression levels of human genes of glutathione metabolism reported in the Human Protein Atlas.** Expression data of genes encoding proteins of glutathione synthesis, reduction and cleavage for astrocytes, and excitatory and inhibitory neurons were obtained from the Human Protein Atlas. Please note that these transcript levels may not accurately reflect those of the corresponding proteins and that no direct correlation of both transcript and protein levels with activity may exist. The abbreviations correspond to γ-glutamylcysteine ligase catalytic (*GCLC*) and modifier (*GCLM*) subunits; glutathione synthetase (*GSS*); glutathione reductase (*GSR*); and γ-glutamyltranspeptidases (*GGT1*, *GGT5*, and *GGT7*).

**Figure 4 ijms-24-08059-f004:**
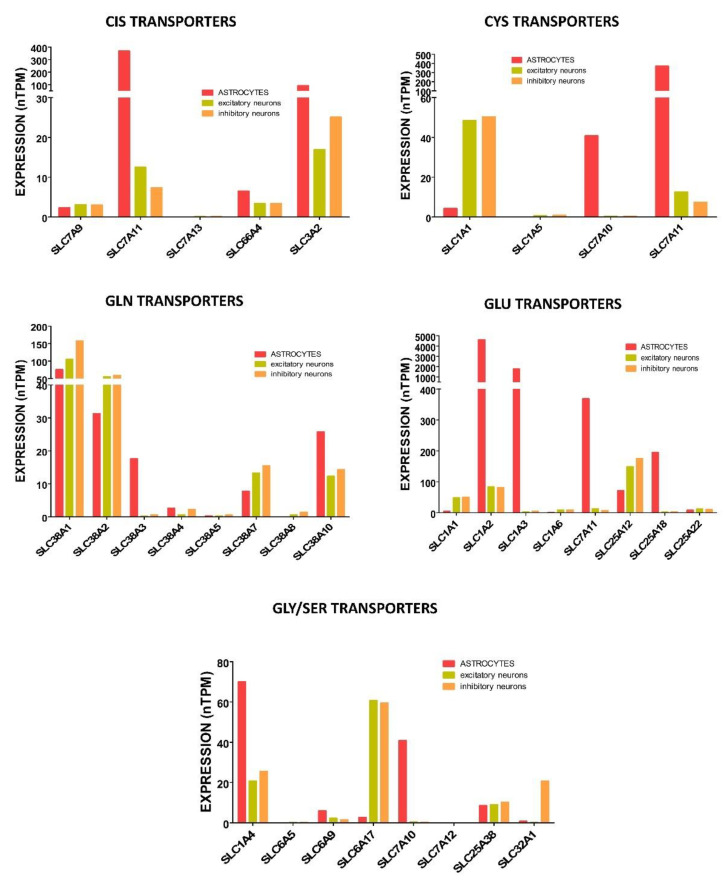
**Expression levels of human amino acid transporters as shown in the Human Protein Atlas.** Data collected from the Human Protein Atlas refer to expression levels of amino acid transporters in astrocytes, excitatory neurons, and inhibitory neurons, although not all transcripts reach significant levels in these cell types. The figure shows a comparison of data obtained from single-cell RNA sequencing (https://www.proteinatlas.org/ (accessed on 14 November 2022)). Both Cis and Cis transporters are expressed in astrocytes and neurons (upper panels), although higher levels for Cis transporters are found in astrocytes, and differences for Cys transporters are less pronounced but still larger for astrocytes. Similar levels of expression for different types of carriers allowing Gly or Ser transport are also reported in this database (lower panel). Expression of Gln transporters is detected both in neurons and astrocytes (left center panel), although levels in neurons are higher. Levels of Glu transporters are extraordinarily high in astrocytes compared to neurons and well above those detected among the amino acid transporters considered in the present work (right center panel). Again, transcript levels found in this database may not reflect the corresponding protein levels. The RNA nomenclature shown in the figure is that found in the database and the encoded proteins/carriers (in parenthesis) are as follows: *SLC7A11* and *SCL3A2* (xCT and 4F2hc subunits of the X_c_^−^ antiporter, respectively); *SLC1A3* (EAAT1); *SLC1A2* (EAAT2); *SLC1A1* (EAAT3); *SLC1A6* (EAAT4); *SLC1A7* (EAAT5); *SCL1A4* (ASCT1); *SCL1A5* (ASCT2); *SLC7A9* (CSNU3); *SLC7A10* (ASC-1); *SLC7A13* (AGT-1); *SLC6A9* (GLYT1); *SLC6A5* (GLYT2); *SLC6A17* (B^o^AT); *SLC25A38* (FLJ20551); *SLC32A1* (Gly export system VIAAT); *SLC38A1* (ATA1) and *SLC38A2* (ATA2); *SLC38A3* (SN1); *SLC38A4* (ATA3); SLC38A5 (SN2); *SLC38A7* (FLJ10815); *SLC25A12* (ARALAR); SLC25A22 (GC1); and *SLC66A4* (lysosomal CTNS Cis carrier).

**Figure 5 ijms-24-08059-f005:**
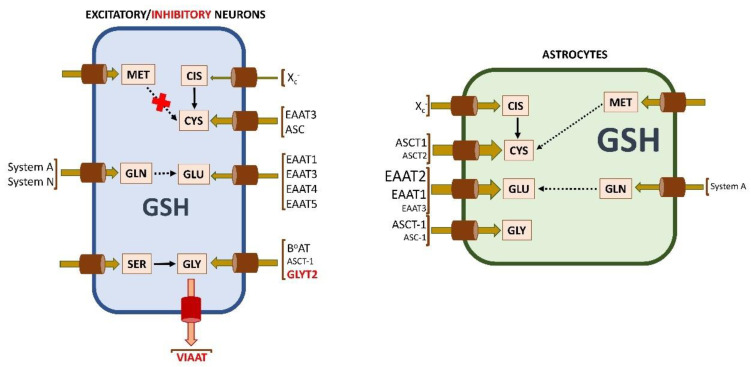
**Schematic representation of the differences in astrocytic and neuronal expression and uptake through amino acid transporters.** Data collected from the Human Protein Atlas and the literature show differences in the expression and activity of transporters involved in the uptake of amino acids required for glutathione synthesis between neurons and astrocytes. These results are summarized in the figure, assuming the existence of a correlation between transcript and protein levels. Abbreviations shown correspond to protein nomenclature and specific carriers found in inhibitory neurons are labeled in red font. The font size attempts to reflect the comparative expression levels of the carriers and the glutathione concentrations, while the width of the arrows simulates the diverse uptake rates estimated in the literature. Discontinuous lines are used to depict the existence of several steps in a specific pathway. The lack of transsulfuration in neurons reported in the literature is indicated by a red cross.

## Data Availability

No new data were created or analyzed in this study. Data sharing is not applicable to this article.

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
