# Peer review of "Appraising the Role of Astrocytes as Suppliers of Neuronal Glutathione Precursors"

_ijms, 2023, doi:10.3390/ijms24098059_

Round 1

Reviewer 1 Report

Here authors review the role of glutathione (GSH) in terms of metabolism, transport, antioxidant capacity and general functions in the neuron-glia circuit, highlighting the anabolic and catabolic enzymes in both compartments. The concept that GSH is the most important astrocytic antioxidant is discussed and updated in the redox environment.

This is an excellent review that covers the last four decades of intense research in the field of GSH and GSSG (the Cys-Cis pair, the changes in GSH/GSSG ratio as a matter of oxidative stress), of how the amino acid precursors are taken up differently by neurons and glia by selective membrane transporters. Authors discuss enzymatic systems and low MW antioxidants in vitro (enriched neuronal and/or astrocytic cultures, mixed or not) and in vivo.

Authors extensively review the varied functions of GSH. The text figures as biochemical schemes are also fine. Only one misspelling error was found (dell-cell line 932). The quality of the text is superior.

I understand that the main message of the manuscript is to discuss how GSH is differently synthesized in astrocytes and neurons and how the compartmentalization could have an impact in the redox scenario. Perhaps one point that the authors missed is that GSH has been raised as a signaling molecule as well. The role of extracellular GSH secreted by transporters or vesicles (discussed from page 19 to the end) could be approached as a messenger interacting with different neuro- and gliotransmitter receptors – in addition to the antioxidant actions. Some authors argue that GSH could even act directly on its own population of receptors (Janáky et al., 1999). Indeed, weakening in GSH function is linked to aging process neuronal loss and neurodegenerative diseases (Iskusnykh, Zakharova, & Pathak, 2022). Glutamate, as the main excitatory transmitter and as a building block of GSH, is particularly related to the glutathione cycle that shapes synaptic activity (Sedlak et al., 2019). This last paper is cited (Ref 177) but discussed in the context of metabolism (line 563) and not of signaling.

Iskusnykh, I. Y., Zakharova, A. A., & Pathak, D. (2022). Glutathione in Brain Disorders and Aging. Molecules, 27(1), 324.

Janáky, R., Ogita, K., Pasqualotto, B. A., Bains, J. S., Oja, S. S., Yoneda, Y., & Shaw, C. A. (1999). Glutathione and Signal Transduction in the Mammalian CNS. J Neurochem, 73(3), 889-902. doi:https://doi.org/10.1046/j.1471-4159.1999.0730889.x

Sedlak, T. W., Paul, B. D., Parker, G. M., Hester, L. D., Snowman, A. M., Taniguchi, Y., . . . Sawa, A. (2019). The glutathione cycle shapes synaptic glutamate activity. Proc Natl Acad Sci U S A, 116(7), 2701-2706. doi:10.1073/pnas.1817885116

Author Response

Please, see the attached file

Reviewer 2 Report

By and large the paper is very well written and clear. However, it would benefit by a native English speaking editor.

The nitrogen isotope experiments mentioned in several places were not based on radiolabeled compounds; N-15 is not radioactive.

The authors rely heavily on RNA data from the Protein Atlas. They should emphasize that mRNA may not accurately reflect levels of protein expression.

In the discussion of extracellular vesicles it could be added that glutathione-dependent enzymes may serve as carriers in the intercellular transport of glutathione. 

Author Response

Please, see the attached file
